

# How many measurements are needed to estimate accurate daily and annual soil respiration fluxes? Analysis using data from a temperate rainforest

Jorge F. Perez-Quezada[1,2*], Carla E. Brito[1], Julián Cabezas[1], Mauricio Galleguillos[1], Juan P. Fuentes[3], Horacio E. Bown[4], Nicolás Franck[5]

[1]Department of Environmental Science and Renewable Natural Resources, University of Chile, Casilla 1004, Santiago, Chile
[2]Institute of Ecology and Biodiversity, Santiago, Chile
[3]Department of Silviculture and Nature Conservation, University of Chile, Casilla 9206, Santiago, Chile
[4]Department of Forest Management and the Environment, University of Chile, Casilla 9206, Santiago, Chile
[5]Department of Agricultural Production, University of Chile, Casilla 1004, Santiago, Chile

*Correspondence to*: Jorge F. Perez-Quezada (jorgepq@uchile.cl)

**Abstract.** Making accurate estimations of daily and annual $R_s$ fluxes is key for understanding the carbon cycle process and
projecting effects of climate change. In this study we used high-frequency sampling (24-measurements per day) of $R_s$ in a temperate rainforest during one year, with the objective of answering the questions of when and how often measurements should be made to obtain accurate estimations of daily and annual $R_s$. In this aim, we randomly selected data to simulate samplings of 1, 2, 4 or 6 measurements per day (distributed either during the whole day or only during daytime) combined with 4, 6, 12, 26 or 52 measurements per year. Based on the comparison of partial-data series with the full-data series, we
estimated the performance of different partial sampling strategies based on bias, precision and accuracy. In the case of annual $R_s$ estimation, we compared the performance of interpolation vs. using non-linear modelling based on soil temperature. The results show that, under our study conditions, sampling twice a day was enough to accurately estimate daily $R_s$ (RMSE<10% of average daily flux), even if both measurements were done during daytime. The highest reduction in RMSE for the estimation of annual $R_s$ was achieved when increasing from 4 to 6 measurements per year, but reductions were
still relevant when further increasing the frequency of sampling. In conclusion, we found that increasing the number of field campaigns was more effective than increasing the number of measurements per day, provided a minimum of two measurements per day was used. Including nighttime measurements significantly reduced the bias and was relevant in reducing the number of field campaigns when a lower level of acceptable error (RMSE<5%) was established. Using non-linear modelling instead of linear interpolation did improve the estimation of annual $R_s$ but not as expected. Given that most
of the studies of $R_s$ use manual sampling techniques and apply only one measurement per day, we suggest making an intensive sampling at the beginning of the study to determine minimum daily and annual frequencies of sampling.



## 1 Introduction

Respiration is the second most important flux in ecosystems after photosynthesis, in terms of the quantities of exchange and the contribution to the total carbon cycle (Schlesinger and Andrews, 2000). Within ecosystem respiration, soil respiration ($R_s$) is considered a key element of the global C cycle, representing about 50-94% of the terrestrial ecosystem respiration,

depending on the period of the year (Curiel Yuste et al., 2005; Goulden et al., 1996). Soil respiration is defined as the aggregation of below-ground processes of heterotrophic (microbial respiration) and autotrophic (root respiration) components (Savage et al., 2009). Accordingly, major differences in $R_s$ are explained by the variation in metabolic activity of both autotrophic and heterotrophic components, which are driven by changes in environmental conditions (Raich and Schlesinger, 1992). This temporal heterogeneity makes the estimations of daily and annual $R_s$ difficult, expensive and time

consuming tasks. Therefore, the development of $R_s$ measurement protocols which maximize the accuracy/measurement frequency ratio for estimating $R_s$ will definitively accelerate the progress of our knowledge about the global carbon balance and its drivers.

Soil respiration has been reported to differ across temporal and spatial scales (Jia et al., 2006; Li et al., 2008; Vargas et al., 2010) as a result of changes in soil temperature (Lloyd and Taylor, 1994; Subke and Bahn, 2010), soil moisture (Bown et al.,

2014; Gaumont-Guay et al., 2006) , vegetation (Bahn et al., 2010; Buchmann, 2000), topography (Kang et al., 2003), soil texture (Dilustro et al., 2005; Pumpanen et al., 2008), and primary productivity (Bahn et al., 2010, 2008; Högberg et al., 2001; Vargas et al., 2011). Among these variables, temperature and soil moisture are the most widely used in empirical prediction models of $R_s$ (Trumbore, 2006). This trend is consistent with the results reported by Chen et al. (2014) who, using a global database, showed that most variation in $R_s$ was explained by mean annual precipitation, closely followed by mean

annual air temperature, soil organic carbon, net primary productivity, pH, tree age, tree height, litter fall biomass, leaf area index, elevation and diameter at breast height.

Soil respiration can be measured with alkali traps or infrared gas analysers (IRGA), the latter being the current reference for $CO_2$ quantification (Davidson et al. 2002). Automatic and manual chamber systems that include IRGA analysers are commonly used to measure $R_s$ and no significant differences between them have been found (Davidson et al., 2002; Irvine

and Law, 2002; Savage and Davidson, 2003). However, these two types of chamber systems do differ in their cost and operational requirements. The use of automated chambers implies a higher equipment cost, allowing higher frequency of measurements at a lower operational cost. In contrast, manually-operated chamber systems are cheaper to buy and allow a higher spatial resolution, but with a higher operational cost. This implies that measurements with the latter type of chambers are usually done with a lower measurement frequency and will less likely include measurements during the night. Based on a

review by Gomez-Casanovas et al. (2013), automated chambers are used to measure $R_s$ in about 24% of the studies, while the rest used manual chambers. In the case of humid forests both approaches are used and, in the case of annual estimates using manual chambers, measurements are usually done only once per day and during daytime (Table 1). Regardless of the



type of sampling and which method is used to estimate the annual flux, little is known about how many measurements should be taken and at what time, in order to obtain more accurate estimates of daily and annual $R_s$.

Temperate forests present high shaded area compared to agricultural and ecosystems with sparser vegetation, such as pasture or shrublands, nevertheless, variations of ±25% of the daily $R_s$ flux has been reported for a temperate mixed hardwood forest,

being the mid-morning measurements the best period to estimate daily mean fluxes (Davidson et al., 1998). However, errors in daily estimation of $R_s$ can be generated if measurements are predominantly made during the warmest part of the day, thus introducing a bias in the estimation (Davidson et al., 2002).

Commonly researchers select times of the day during the morning, to get the estimates of daily $R_s$ fluxes. For example, Tang et al. (2006) suggested that measurements taken at 9:00 AM where representative of the daily mean flux in subtropical

forests, however this value was calculated on the basis of 10 measurements made during the morning. This idea was tested by Qin and Yang (2013) using ecosystem respiration data which demonstrated that measurements taken at 9:00 AM were significantly higher compared with the daily mean. Similarly, Davidson et al. (1998) found that their measurements made between 9:00 AM and 12:00 PM adequately represented the average daily $R_s$ flux in a temperate mixed hardwood forest, although this conclusion was derived from intensive measurements made on only two consecutive days.

Based on the estimation of seasonal or annual $R_s$, Savage et al., (2008) developed a protocol for data quality assurance in a mixed hardwood forest and determined that a sampling strategy with a bi-weekly frequency would be optimal. However, sampling was done only between 9:00 AM and 3:00 PM. Gomez-Casanovas et al. (2013) used high-frequency data to estimate the performance of different gap filling techniques to estimate annual $R_s$ in experimental plots. As expected, they found that increasing the data gap fraction decreased the ability of all gap models to accurately predict $R_s$ (above 15%

decrease) and increased the variability of the prediction.

Using a high frequency sampling scheme (24 measurements per day) during one year in a temperate rainforest, we aimed to answer the still open questions of when and how many measurements per day and per year should be performed in order to adequately estimate $R_s$ fluxes. Assessing performance of estimators based on their bias, precision and accuracy, as proposed by Walther and Moore (2005), our objectives were: (i) to assess the performance of estimating daily $R_s$ fluxes based on

different number of measurements per day, (ii) to compare the performance of estimating the annual $R_s$ flux using linear interpolation or modelling based on different number of measurements per year, and (iii) to analyse the effect of including night-time measurements on the accuracy of the estimation of daily and annual $R_s$.



## 2 Methods

### 2.1 Study site

The study was carried out in a temperate rainforest at the Senda Darwin Biological Station (Carmona et al., 2010), a Long Term Socio-Ecological Research site located 15 km east of Ancud, in the Chiloé Island, Chile (41° 52' S, 73° 40' W)

(Figure 1). The dominant species are large emergent trees (up to 25 m) of *Drimys winteri*, *Podocarpus nubigena*, *Nothofagus nitida*, and *Saxegothaea conspicua*, while the understory species are seedlings and saplings of the dominant trees and some tree species of shrub habit, such as *Tepualia stipularis*. Tree trunks and fallen logs are covered by several bryophyte species, including mosses and liverworts. Soils are generally thin (<1 m), originated from Pleistocenic moraine fields and glacial outwash plains, with often poor drainage (Aravena et al., 2002).  Soils are acidic (pH 3.9±0.4), with very low bulk density

(0.2±0.04 g cm$^{-3}$), high total C (39±9% dry weight) and low total N (1.3±0.2% dry weight) (Aravena et al., 2002).  The climate is temperate with a strong oceanic influence. Meteorological records (1997-2008) at the Senda Darwin Station indicate an annual average temperature of 10°C, with a maximum average of 16°C in January and a minimum average of 5°C in July. Annual rainfall is 2.000-2.500 mm, with an average of 2.110 mm and a dry period during January-February. The year of the study (August 2013-July 2014) was wetter (2.383 mm) and cooler (9.4°C) than average, although the summer

months (December-February) were drier and hotter (Figure 2).

### 2.2 Automated measurements of $R_s$ fluxes and environmental variables

The variability of forest conditions was preliminary assessed in terms of canopy cover and other stand related parameters. According to this, three soil respiration chambers were installed to cover the range of these variables, which were assessed in 3-m radius plots around each chamber. Table 2 shows the basic statistical parameters of the forest stand, soil and annual $R_s$

flux.

$R_s$ fluxes were measured with an automated soil $CO_2$ flux system (model LI-8100, LI-COR, Lincoln, Nebraska, USA), connected to a multiplexer (model LI-8150, LI-COR) and three 20-cm diameter closed chambers (model LI-8100-104, LI-COR). The chambers were installed over a PVC collar buried into the soil, which stayed in place during the whole sampling period and were kept free of photosynthetically active material. Measurements were done for 2 minutes on each chamber

every hour, for one year, starting in August 8, 2013. Considering power supply problems that occurred on 6 days during the sampling period, the total data set was 25,776 records (24 measurements per day × 358 days × 3 chambers).

Soil temperature ($T_s$) and soil water content ($\Theta$) were monitored at 5 cm depth, close to each chamber, using thermocouple probes (model TCAV, Campbell Scientific Inc. (CSI), Logan, UT, USA) and water content reflectometers (model CS616, CSI). Data were collected with a datalogger (model CR3000, CSI) every 30 minutes.



### 2.3 Generation of partial-data series for estimating daily $R_s$ fluxes

For analysing the effects of making a different number of measurements per day and including or not night-time measurements on the performance of daily $R_s$ flux estimations, we generated partial-data series. For this, we established four premises:

1) The average from the three soil chambers represented the forest $R_s$ flux.

    2) The average of the 24 measurements made in one day (00:00 to 23:00) was considered to be the best estimate of daily $R_s$ flux,

    3) Daily sampling was made at frequencies of 1, 2, 4 or 6 measurements per day, which were averaged to estimate the partial-data daily $R_s$ flux.

4) Two sampling types were defined, considering all the measurements made during one day (*day-night*), or considering measurements made only during daytime (07:00 to 19:00, *day*).

The partial-data series were then generated by randomly selecting measurements. This process was different for the two sampling types defined in 4):

    a. For the day-night sampling, after randomly selecting one day (out of the 358 possibilities), the initial time

15        of sampling was also randomly selected (out of the 24 possibilities); the other time(s) of measurement was selected equidistantly from this initial value, maximizing the time distance between samplings to fit the number of measurements in one day. In the case of 1 measurement per day, the time of measurement was the same as the initial time of sampling.

    b. For the day sampling, to maintain the number of measurements per day and to make them as equidistant as

20        possible, the time of measurements were randomly selected from windows of time, as shown in Figure 3.

### 2.4 Generation of partial-data series for estimating the annual $R_s$ flux

We defined different frequencies of sampling assuming that the most common sampling schemes are seasonal (summer, autumn, winter and spring), bimonthly, monthly, biweekly or weekly. These frequencies implied 4, 6, 12, 26 and 52

measurements per year, respectively, which represented our partial-data series. The best estimate of the annual $R_s$ flux was calculated considering all available data. Because the error in estimating the annual $R_s$ flux is not independent of the number of measurements during one day, we combined the daily and annual frequencies of sampling. The day and day-night sampling types were also considered for the estimation of annual $R_s$ flux using partial sampling. The days of measurements were determined by selecting a random initial day of measurement and then adding (or subtracting) the maximum possible

time distance to fit the number of measurements within a year. In addition, a buffer range of days was added around the days originally selected to simulate the fact that, in most field studies, the real day of measurement is not exactly the planned one.




This buffer range varied with the frequency of measurement, being ± 16, 8, 4, 2 and 1 day, for the frequencies of 4, 6, 12, 26 and 52 measurements per year, respectively.

Once the selection of daily and annual measurements was done, we used two different approaches to estimate the annual $R_s$ flux. The first approach was linearly interpolating the daily fluxes, while the second one was modelling based on

environmental variables. For selecting the model, we used all the available data and found that the non-linear model based on $T_s$ (Lloyd and Taylor, 1994) was the most appropriate for our data set ($R_s = 0.744$ e$^{(0.13 * Ts)}$; $R^2=0.89$) : although soil water content did show a significant negative linear relation with $R_s$, adding this variable to the model did not improve the adjusted coefficient of determination, so the simplest model (based only on $T_s$) was preferred.

**2.5 Performance of estimations of daily and annual $R_s$ fluxes using different sampling frequencies**

For estimating the performance of the estimations of $R_s$, we used unscaled measures of bias, precision and accuracy, according to Walther and Moore (2005). Bias is defined as the deviation of measurements from the mean, which is usually due to faulty measuring devices or procedures. Bias therefore leads to either underestimation or overestimation of the true value. Precision is the statistical variability of an estimation procedure and is considered to be independent of the true value.

Finally, accuracy defines the overall performance of an estimator and is the combination of bias and precision.

All three parameters were calculated for each sampling frequency based on 10,000 partial-data series, which were generated as described in sections 2.3 and 2.4. The Bias of the daily estimation for each frequency of sampling was calculated as:

$$Bias = \sum_{\substack{1 \le i \le m \\ 1 < j < n}} Rs_{ij} - Rsb_{ij} \,, \tag{1}$$

where $Rs_{ij}$ is the partial-data estimation of daily $Rs$ flux and $Rsb_{ij}$ is the best estimate of the daily $R_s$ flux, for sample $i$

(m=10,000 iterations) and day $j$ (n=358).

The precision was estimated as the standard deviation (SD) of the partial-data series estimations of the daily flux, using the values selected in Equation 1:

$$SD = \left( \frac{\sum_{\substack{1 \le i \le m \\ 1 < j < n}} (Rs_{ij} - \overline{Rsb_{ij}})^2}{m} \right)^{1/2} \tag{2}$$

Finally, the accuracy of the estimation of daily $R_s$ for each frequency of sampling was estimated as the root mean square

error (RMSE), which was calculated as:

$$RMSE = \left( \frac{\sum_{\substack{1 \le i \le m \\ 1 < j < n}} (Rs_{ij} - Rsb_{ij})^2}{m} \right)^{1/2} \tag{3}$$

All hourly values and days had the same chance of being selected. For estimating the performance of estimations of annual $R_s$ flux, the same three statistical parameters (Equations 1-3) were used, considering as best estimate the annual $R_s$ flux calculated using all available data. All the statistical analyses were done using the software R, version 3.1.2 (R Core Team,

30 2014).





## 3  Results and Discussion

### 3.1. Daily and annual $R_s$ fluxes and environmental variables

The daily $R_s$ values showed low variability between chambers located under different tree cover in cold months (June-September) with values around 1.5 μmol m$^{-2}$ s$^{-1}$ (Figure 4-A). Soil respiration increases up to 6 μmol m$^{-2}$ s$^{-1}$ in mid-January, showing a wider spread between chambers during warmer months (Figure 4-A).

The same pattern was observed for $T_s$, which moved from values around 6 ºC during the winter, up to a maximum of 16 ºC in the middle of the summer (Figure 4-B). The opposite pattern was observed in $\Theta$, where the highest and very stable values were observed during the winter months around 0.55 and decreased during the summer to around 0.4 (Figure 4-C). A greater variability between chambers was observed in $\Theta$ compared to $T_s$ (Figures 4-B and 4-C).

The difference in $R_s$ between winter and summer periods may have been increased by the fact that during the study year, winter was wetter and cooler than average, while the summer months were drier and hotter (Figure 2). This situation matches the prediction of climate change in this area for the period 2071-2100 reported by Fuenzalida et al., (2007). If this dryer summer conditions continue, a reduction is expected for evapotranspiration (15%) and aboveground biomass (27%) in this ecosystem type (Gutiérrez et al., 2014).

### 3.2. Effects of sampling frequency and nighttime measurements on daily estimations of $R_s$

In the scenario where only daytime measurements were considered, the bias was always positive, around 0.35 g $CO_2$ m$^{-2}$ day$^{-1}$, meaning an overestimation of daily $R_s$ fluxes (Figure 5-A), which coincides with the suggestions of Davidson et al. (2002) and Qin and Yi (2013). Increasing the number of measurements in this scenario from 1 to 6 per day decreased the bias only slightly. Including night-time measurements decreased the bias for estimating daily $R_s$ fluxes to nearly zero (Figure 5-A), with little difference observed when making 2, 4 or 6 measurements per day.

The precision (SD) of the daytime scenario was around 4.2 g $CO_2$ m$^{-2}$ day$^{-1}$, showing only a small decrease when adding more measurements during the day (Figure 5-B). Adding measurements during the night decreased the SD only 5% to a value of 3.99 g $CO_2$ m$^{-2}$ day$^{-1}$.

The RMSE for only daytime measurements showed an important decrease when comparing the frequencies of 1 and 2 measurements per day (1.45 and 0.82 g $CO_2$ m$^{-2}$ day$^{-1}$, respectively) (Figure 5-C), while still decreasing but less with 4 and 6 measurements per day. Adding nighttime measurements decreased the RMSE considerably, especially for sampling frequencies ≥2, which had a mean RMSE of 0.39 g $CO_2$ m$^{-2}$ day$^{-1}$.

The precision of daily measures of $R_s$ was one order of magnitude higher than the bias, when comparing only daytime measurements, difference that was much larger when considering day and night measurements (Figures 5-A and B). According to our results, the best option was to measure twice during the day, even if both measurements were restricted to




daytime, because this frequency yields an RMSE <10% of the mean daily value (Figure 5-C). If the accuracy threshold was to be set at 5%, the minimum frequency of sampling required was 4 times per day, including night-time measurements.

**3.3 Effect of frequency of sampling and inclusion of night-time measurements on annual estimations of $R_s$**

Figure 6 shows the statistical parameters for the annual estimations of $R_s$ using both linear interpolation and non-linear

modelling approaches. This figure does not include the results of 6 measurements per day because they were almost identical to measuring 4 times per day. All the results are shown in Supplementary material Table S1. In the scenario of only daytime measurements, the bias was always positive around 0.12 kg $CO_2$ m$^{-2}$ year$^{-1}$ for all frequencies of daily sampling (Figure 6-A). Adding nighttime measurements made the bias negative, but closer to zero (Figure 6-B). Only in the latter case using modelling, instead of interpolation, performed better, i.e. showed a bias closer to zero.

In summary, regardless of the annual frequency of sampling, making measurements only during daytime represented a positive bias (overestimation) of the annual $R_s$ flux. Sampling also during nighttime practically eliminated the bias. However, this does not mean that error does not exist, but only that this error occurs equally above and below the mean. The error (defined as the observed – modeled values) decreases greatly when moving from sampling 4 to 6 per year (Supplementary material Figure S1).

The SD of only daytime measurements moved from a maximum around 0.52 kg $CO_2$ m$^{-2}$ y$^{-1}$ for the frequency of sampling 4 days per year and measuring once per day, to a minimum 0.07 kg $CO_2$ m$^{-2}$ y$^{-1}$ for the frequency of 52 per year and 4 per day (Figure 6-C). There was an improvement (lower SD) when adding nighttime measurements (Figure 6-D) and more so when using a modelling approach to estimate the annual $R_s$ flux. This latter improvement was more noticeable when sampling 2 or 4 times per day and 12 times per year.

Because the parameter we used to represent the precision of the annual estimation (SD) accumulates the difference between observed and modeled values, the magnitude of the error associated to precision was much larger than the bias, making the value of precision almost identical to the accuracy parameter (Figures 6-C and -D compared to 6-E and -F).

The RMSE of the daytime scenario showed a decreasing trend when increasing the frequency of sampling from 4 to 6, 12, 26 and 52 days per year, with mean values of 0.42, 0.34, 0.26, 0.22 and 0.20 kg $CO_2$ m$^{-2}$ y$^{-1}$, respectively (Figures 6-E and -

F). Within each annual frequency of sampling, there was a considerable decrease in the RMSE when comparing the frequency 1 per day with the frequencies 2 and 4 measurements per day. In the day-night sampling (Figure 6-F), the RMSE decreased between 0.01 - 0.11 kg $CO_2$ m$^{-2}$ y$^{-1}$ compared to the corresponding partial-data series of daytime sampling. The non-linear modelling approach performed better than linear interpolation only when sampling was performed day-night (Figure 6-F). When sampling only during the day (Figure 6-E), the modelling approach was clearly better only when

sampling 2 or 4 times per day and 12 times per year.





If sampling was done only once a day, sampling once a month was the minimum frequency required for obtaining accurate estimates of annual $R_s$ (RMSE <10% of the annual flux) (Figures 6-E and -F). When establishing a threshold of accuracy of 5%, for sampling only during daytime, a minimum of 2 measurements per day, and biweekly field campaigns were required (Figure 6-E). For the latter accuracy threshold, if nighttime measurements were included, sampling twice a day was also

required but only at a monthly interval (Figure 6-F). This example highlights the importance of performing nighttime measurements.

The effect of sampling more times per day on the error of annual $R_s$ estimation was more dramatic when moving from 1 to 2 measurements per day, less so when moving from 2 to 4 and almost negligible when moving from 4 to 6 (Figures 6-E and -F). On the contrary, increasing the number of field campaigns (days sampled per year) showed a continuous and almost

linear decrease when moving from 4 to 52 days per year (Figures 6-E and -F).

### 3.4 Other sources of error and means of improving the accuracy of $R_s$ estimations

Table 1 shows that there is great variability in both daily and annual frequency of sampling in studies that measured $R_s$ in humid forests. Most studies that used manual chambers sampled only once per day, only during daytime and between 6 and 48 times per year. According to our results, only the estimations from studies that sampled annually ≥12 times would yield

an RMSE <10% of the annual flux.

We tested both linear interpolation and modelling based on soil temperature as gap filling approaches, expecting that the latter would yield lower error in the annual $R_s$ flux, as suggested by several authors (Davidson et al., 1998; Raich and Schlesinger, 1992; Wang et al., 2006). However, the modelling approach represented a clear decrease in error only when sampling 2 or 4 measurements per day and once a month, which in our study represented an intermediate number of total

measurements per year. This makes sense, given that fitting a non-linear model requires a minimum number of data, over which modelling performs almost equal than interpolation. Gomez-Casanovas (2013) compared nine different gap-filling methods and concluded that the linear interpolation method was the second best-performing method, while the method based on $T_s$ was among the most poorly-performing methods. Here we found that modelling based on $T_s$ was a better method, particularly in reducing the bias, but not as expected.

Unfortunately, we cannot compare our results with the studies summarized in Table 1, because no information is given about the level of accuracy of the daily or annual estimations of $R_s$ flux. Because our study site is very close to the ocean, daily and annual climatic variability are low. We expect that obtaining good performing estimates of daily and annual $R_s$ fluxes under more variable environmental conditions would require more frequent sampling. This is expected not only because of the larger diurnal and annual oscillations of soil temperature and humidity, but also because of the higher variability in

biological activity of trees during the year.

Finally, we agree with Gomez-Casanovas et al. (2013), in relation to the need of improving and standardizing the techniques to estimate the annual $R_s$ for understanding its role in the global C cycle. According to our results, part of this standardization



process should not only include the gap-filling approach, but also the frequency of daily and annual sampling. Accomplishing this will require studies similar to ours under different environmental conditions.

## 4 Conclusions

According to our observations in a temperate rain forest site, if the research question seeks accurate daily estimations of $R_s$, sampling $\geq 2$ per day would be necessary to obtain an accuracy that represents an RMSE <10%. Adding nighttime measurements improved the accuracy and precision slightly and, most importantly, decreased the bias, which was always positive when sampling only during daytime.

In general, accuracy of most combinations of daily and annual sampling frequencies used for modelling annual $R_s$ achieved
high accuracy (RMSE<10%). The reduction in RMSE was highest when increasing measurements from 4 to 6 per year, but was still relevant when further increasing annual measurement frequency. We therefore recommend increasing annual rather than daily measurement frequency and including a minimum of one daytime and one night-time measurement. Actually, in the case of establishing a high accuracy threshold (RMSE<5%), making one of the two measurements during the night in one day, decreased the number of field campaigns per year from 26 to 12.

The decrease in error when using modelling instead of linear interpolation for estimating $R_s$ annual flux was evident only for intermediate sampling frequency levels which, in our case, was represented by doing 2 or 4 measurements per day and field campaigns once a month.

As a general measure for reducing the errors originated from partial sampling of $R_s$ during one day and during the whole year, we recommend making an intensive sampling (including nighttime measurements) at the beginning of the study. This
should allow determining the best time(s) and the minimum frequency for sampling. We expect that this process may be more critical where environmental conditions are more variable compared to the conditions in our study site.

### Acknowledgements

The authors thank the funding from the National Commission for Scientific & Technological Research of Chile (grants
FONDEQUIP AIC-37 and FONDECYT 1130935). They also thank the administration and personnel at the Senda Darwin Biological Station and Dr. Richard Plant for his valuable comments on a preliminary version of the manuscript.





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



**Table 1. Sampling schemes of studies where annual $R_s$ was estimated in humid forests.**

| Forest type (country) | Period | Sampling time | Daily freq. | Annual freq. | Type of sampling | Method for estimating annual flux (variables used)*** | Reference |
|---|---|---|---|---|---|---|---|
| North temperate forests (USA) | 32 | - | 1 | 9** | A | Linear interpolation | Fisk et al. (2004) |
| Temperate mixed-hardwood forest (Korea) | 8 | - | 1 | 9 | M | Non-linear model ($T_s$) | Kang et al. (2003) |
| Temperate beech forest (France) | 18 | 7:30 - 16:00 | 1 | 12 - 24 | M | Non-linear model ($T_s$, $\Theta$) | Ngao et al. (2012) |
| Coniferous forest (China) | 12 | 10:00 - 12:00 | 1 | 24 | M | Non-linear model ($T_s$) | Xu et al. (2015) |
| Mixed hardwood old-growth forest (USA) | 12 | 9:00 - 15:00 | 1 | 25 | M | Linear interpolation | Savage et al. (2008) |
| Mixed hardwood forest (USA) | 12 | 9:00 - 13:00 | 1* | 28 | M | Linear interpolation and Non-linear model ($T_s$) | Davidson et al. (1998) |
| Mixed and broadleaved forest (China) | 12 | 09:00 | 1 | 48 | M | Linear interpolation | Tang et al. (2006) |
| Temperate mixed forest (Belgium) | 12 | - | 2 | 12 | M | Non-linear model ($T_s$, $\Theta$) | Curiel Yuste et al. (2005) |
| Subtropical forest (China) | 12 | 9:00 - 12:00 | 4 | 48 | M | Non-linear model ($T_s$) | Yan et al. (2006) |
| Subtropical forest (China) | 12 | 0:00 - 23:00 | 24 | 12 | A | Non-linear model ($T_s$) | Yan et al. (2006) |
| Cool-temperate deciduous forest (Japan) | 36 | 0:00 - 23:00 | 24-48 | 12 - 24 | A | BGC-model ($T_s$, $T_a$, Pp, VPD) | Kondo et al. (2015) |
| Cool-temperate deciduous forest (Japan) | 48 | 0:00 - 23:00 | 24-48 | 12 - 24 | A | Non-linear model ($T_s$) | Mo et al. (2005) |
| Mixed hardwood old-growth forest (USA) | 12 | 0:00 - 23:00 | 48 | 180 | A | Linear interpolation | Savage et al. (2008) |

* Assumed to be 1 because no information was reported. **Winter months were estimated from a different study.

*** Abbreviations are: $T_s$, soil temperature; $\Theta$, soil water content; $T_a$, air temperature; Pp, precipitation; VPD, vapor pressure deficit.



**Table 2. Characteristics of the forest stand, soil (30 cm) and annual $R_s$ flux (n=3).**

| Attribute | Mean | SE | Min | Max |
|---|---|---|---|---|
| Soil bulk density (g cm$^{-3}$) | 0.49 | 0.2 | 0.16 | 0.85 |
| Soil carbon content (%) | 40.16 | 1.65 | 19.24 | 54.10 |
| Total litter biomass (Mg ha$^{-1}$) | 18.3 | 2.8 | 6.4 | 28.6 |
| Litter biomass Oi layer (Mg ha$^{-1}$) | 6.4 | 0.9 | 1.9 | 10.9 |
| Litter biomass Oe layer (Mg ha$^{-1}$) | 11.9 | 2.2 | 3.2 | 18.8 |
| Total root mass (Mg ha$^{-1}$) | 23.6 | 6.5 | 7.0 | 70.1 |
| Coarse root mass (Mg ha$^{-1}$) | 16.4 | 5.8 | 3.5 | 58.9 |
| Fine root mass (Mg ha$^{-1}$) | 7.1 | 0.9 | 3.3 | 11.1 |
| Canopy cover (%) | 84.3 | 7.35 | 69.7 | 92.95 |
| DBH (cm) | 11.13 | 9.44 | 5 | 65 |
| Density (trees ha$^{-1}$) | 1521 | 623 | 354 | 2475 |
| Total basal area (m$^2$ ha$^{-1}$) | 44.68 | 41.69 | 1.13 | 128.04 |
| Annual $R_s$ flux (kg CO$_2$ m$^{-2}$ year$^{-1}$)* | 4.15 | 0.31 | 3.81 | 4.76 |

\* The true $R_s$ flux was estimated using the whole-data series.





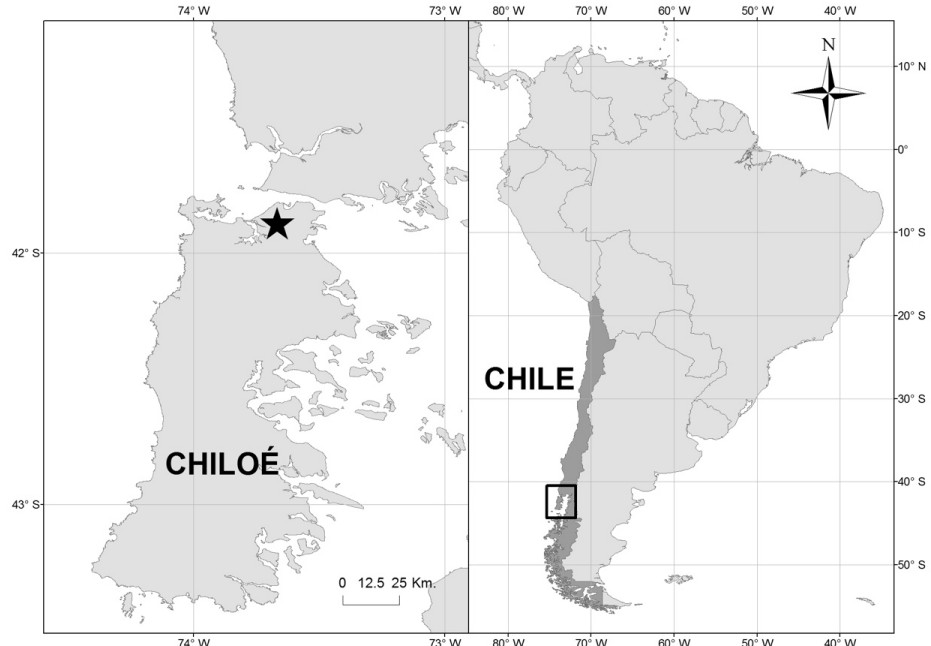

**Figure 1: Location of the Senda Darwin Biological Station (marked with a star) at the Chiloé island.**




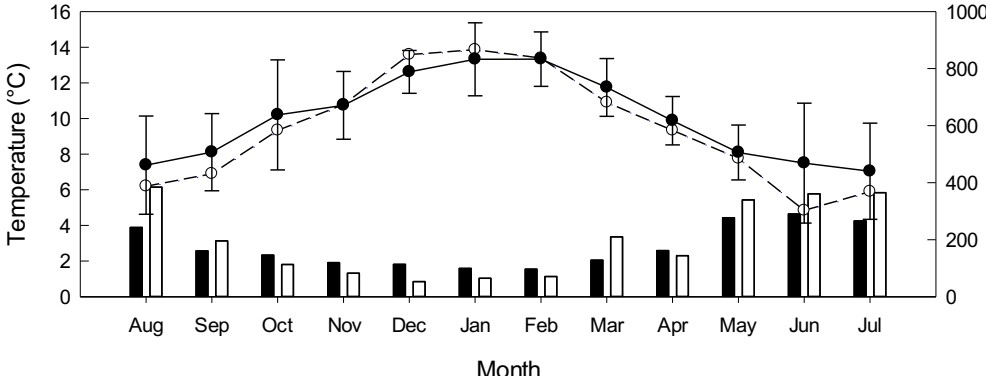

**Figure 2: Mean long-term (1999–2012) (black) and August 2013-August 2014 (white) monthly precipitation (bars) and air temperature (circles).**





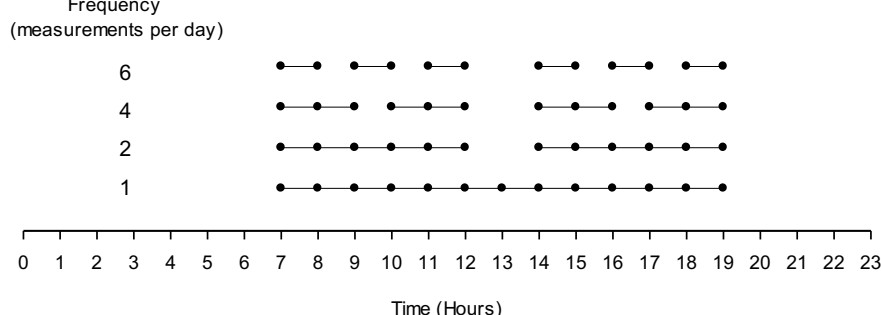

**Figure 3. Windows of time for randomly selecting measurements of different sampling frequencies, for the only day-time sampling. The lines represent the windows of time and the dots represent the exact time of measurements.**




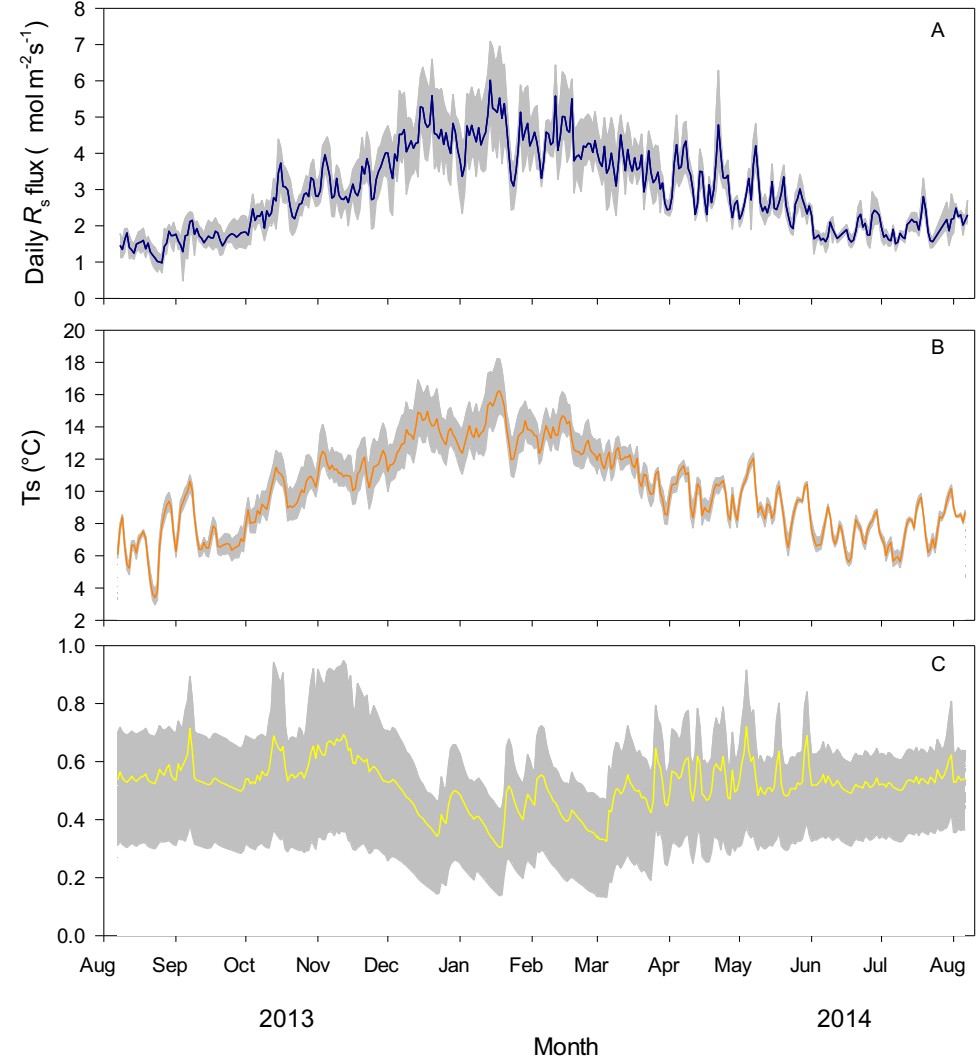

**Figure 4. Daily mean $R_s$ (A), soil temperature (B) and soil water content (C). Shaded area represent the minimum and maximum range of three sampling points.**



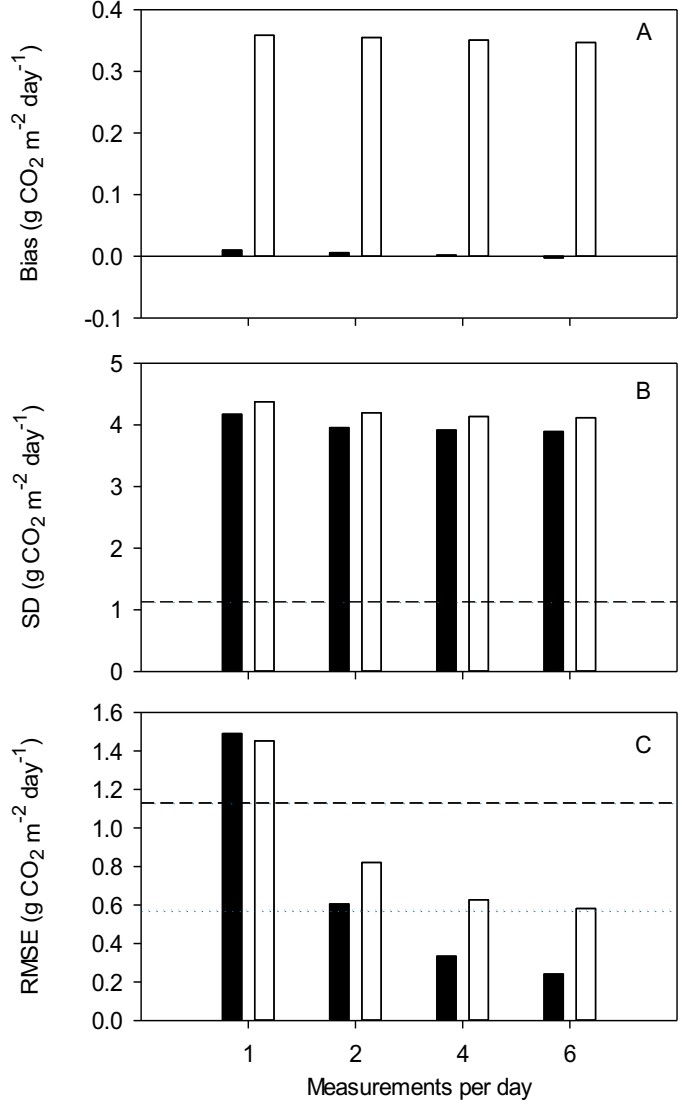

**Figure 5. Statistics of the estimation of daily $R_s$ flux from partial-data series, considering day and night measurements (dark bars) and only daytime measurements (light bars) ($n$=10,000). Statistical parameters are bias (A), standard deviation (B) and root mean square error RMSE (C). The reference lines in panels B and C represent 10% (dashed)**

5    **and 5% (dotted) of the mean daily $R_s$ flux (11.38 g $CO_2$ m$^{-2}$ day$^{-1}$).**





**Figure 6. Bias (A, B), SD (C, D) and RMSE (E, F) of partial-data series, related to both daily and annual frequencies of sampling, considering only daytime measurements (left), or day-night measurements (right). Symbols represent the daily frequency of sampling: red circle, 1; blue triangle, 2; green square, 4. Frequency of 6 measurements per day was not included to make the graph clearer, because the error values did not differ much from the ones for 4**

5 **measurements per day. The approach to estimate the annual flux are differentiated as follows: blank symbol and pointed line are linear interpolations; solid symbol and solid line are non-linear modelling. The reference lines in C, D, E and F panels represent 10% (dashed) and 5% (dotted) of the annual $R_s$ flux (4.15 kg $CO_2$ $m^{-2}$ $y^{-1}$).**