# Peer review of "How many measurements are needed to estimate accurate daily and annual soil respiration fluxes? Analysis using data from a temperate rainforest"

_Biogeosciences, 2016_

## Short Comment (SC1) · 16 Sep 2016

The lack of further analysis of regular cycles such as the diurnal cycle behavior of soil respiration makes this work be incomplete. For example, are maximum $CO_2$ emissions concentrated in some period of the day as generally happens with carbon fluxes?. This would allow better evaluation of the suitability of the sampling strategy.

---

## Author Comment (AC1) · 21 Sep 2016

Response to short comment by Simon Lindemann: The diurnal cycle pattern in our study site shows higher soil respiration values during daytime (18.2% higher compared to nighttime values). We do not explicitly mention this very typical pattern, but the Bias of the flux estimations being always positive for the only-daytime sampling suggests this idea. The annual cycle behavior of soil respiration is presented in Figure 4-A. The effect of diurnal and annual variability of environmental variables on the frequency of sampling required for obtaining good performing estimates of soil respiration is dis-

cussed in page 9 (line 26).

---

## Referee Comment (RC1) · Anonymous Referee #1 · 28 Sep 2016

General Comments This paper is well written and presented. The specific questions the authors address are 1) to assess the performance and accuracy of different numbers of measurements per day 2) compare the performance of linear vs non-linear gap filling based on sampling frequency 3) analyze the effects of including night time respiration measurements on estimations of daily and annual respiration. I disagree with the authors that questions still remain on how many samples are needed per day; this has been addressed in the literature, which the authors cite. I do agree with the authors that how many samples per year as well as gap filling techniques are issues that need addressed. The authors presented a good examination of issues related to sampling

frequencies, gap filling strategies and methods for assessing their performance. This manuscript is very useful to researchers planning an effective sampling strategy and contributes to the overall understanding of estimates of daily and annual respiration.

I have a couple of issues that the authors should address:

1. The authors used only 3 chambers to represent the "true" soil respiration from their site. In the Davidson et al. 2002 paper that the authors cite, they have a table showing how many soil respiration measurements are needed to be within a certain % of the "true" population mean in a northern temperate forest. Using this as a guideline, the authors, having used only 3 chamber measurements at their site, may only be within $\pm 50\%$ of the true population mean. This is something that I think the authors should address in their discussion. Their manuscript is intended to give guidance as to sampling frequency and so they should also reference how many samples may be necessary to capture the "true" respiration mean per site.

2. I understand the use of only the soil temperature model for gap filling since the authors did not see an increase in model fit when adding soil moisture. However, my concern is that there are issues with the soil moisture measurements, the wide range in soil moisture measurements among the 3 respiration chambers is somewhat suspicious. Not including moisture in the gap filling model may have changed the outcome of non-linear gap filling strategy. Can the authors comment on their moisture measurements and the potential impact on their gap filling results? There are also questions regarding the soil moisture measurements below.

Scientific Questions

Pg 4 line 29: Did the authors conducted a soil specific calibration for the CS616 probes or use the supplied calibration equation? The bulk density of the soil shows a very wide range in Table 2 and these types of probes do not function as well in soils with low bulk density. Further Figure 4c graph shows a very large range of measured soil water contents among the 3 probes, this might be more related to the calibration equation

used than to a true range of soil moisture at the site.

Pg 6 line 6: The equation presented here is the Van't Hoff equation: although cited in the Lloyd and Taylor 1994 paper, it is not their equation.

Pg 7 line 3: The authors talk about low variability in the cold month and higher in warmer. Can the authors add estimates of the coefficient of variation for these periods?

Figure 4a: The authors use Rs in $\mu$mols m-2 s-1 in this graph; but use mg in other graphs. It would be preferential to use one type of unit throughout the manuscript. Also note that Figure 4a is missing the $\mu$ in the y axis label.

Please add the units for soil water content-. these are missing on graphs and in the text.

Technical Correction

Figure 2: please label the secondary y axis.

Figure 4c: please label the y axis

---

## Referee Comment (RC2) · Anonymous Referee #2 · 11 Nov 2016

The topics raised in this paper are very relevant and informative for researchers planning soil respiration measurements. When comparing CO2 fluxes among different studies it is essential that a standard protocol for measuring daily and annual CO2 is implemented. This manuscript is well written and tackles some of the uncertainties and questions that many researchers encounter when planning field work/ field campaigns. Using an extensive data set, the authors address these specific questions: 1) to assess the performance of estimating daily Rs fluxes based on different numbers of measurements per day 2) to compare the performance of estimating the annual Rs flux using linear vs non-linear interpolation or modelling based on different number of

measurements per year and 3) to analyse the effects of including night time respiration measurements on the accuracy of the estimations of daily and annual Rs.

There are a couple of points that should be given some attention:

1) I agree that the issue of daily measurements is still an open question. You mention that most studies take samples in the morning. Could you elaborate for the reader as to why most studies have chosen this time? Also, you conclude that the more appropriate number of samples taken for an effective sampling strategy should be a minimum of two samples per day (one day-time and one night-time). I realise that you propose an assessment at the beginning of each study but could you give a general approximation (time window) as to when those two samples should be taken? For example, should they be taken between 7:00-12:00 and 19:00-23:00?

2) Only 3 chamber measurements were used for this study. Although you are not specifically looking at the spatial variation, I think some discussion needs to be given on the number of samples necessary when advising other researchers on sampling protocols.

Specific science and written points:

Pg 3 line 9: 'were' instead of 'where' Figure 2 and 4: Simply reiterating the previous reviewer. Make sure the y axis and secondary y axis are labelled correctly and include the soil water units in the text and graphs.
* * *

---

## Author Response (AR1)

Dear Dr. Subke,

Please find below the responses to all the comments we received about our manuscript. We used blue font both in our responses and in the changes we implemented in the text.

Best regards,

Jorge F. Perez-Quezada

**Responses to Reviewers comments**

**Comments by Associate Editor (Dr. Jens-Arne Subke)**

Comments to the Author:

Dear Dr Perez-Quezada

Many thanks for your responses to comments made in the interactive discussion. All referees are supportive of your manuscript, and have made helpful comments to improve clarity of your main message, and detailed corrections. Your responses to referees adequately address these requests, and I'd like to ask you to implement the proposed changes in your manuscript for a final assessment.

R: We have responded and implemented all the changes derived from the reviewers' comments.

It is clear from all referee comments that the issue of capturing diurnal variability remains a key challenge for field studies. Your proposed changes are welcome in further clarifying this. I think that you are right in stressing that there may not be a "right" window that should be used al every site, and the point that knowledge of diurnal variability and a targeted sampling strategy that takes account of this is required should be made carefully.

R: Yes, we decided not to modify the text because our recommendation of making an intensive sampling at the beginning of the study covers all these issues.

Your preference for using mass rather than molar units for annual respiration fluxes is fine, but please clarify whether you refer to grams of C or grams of $CO_2$ in the figure. Using molar units avoids this problem, but when stating specifically what the mass refers to, this is acceptable.

R: We have added "$CO_2$" in the text and figures to avoid confusion about the units.

With best regards,

Jens-Arne Subke

**Responses to short comment by Simon Lindemann**

The lack of further analysis of regular cycles such as the diurnal cycle behavior of soil respiration makes this work be incomplete. For example, are maximum $CO_2$ emissions concentrated in some period of the day as generally happens with carbon fluxes?. This would allow better evaluation of the suitability of the sampling strategy.

R: The diurnal cycle pattern in our study site shows higher soil respiration values during daytime (18.2% higher compared to nighttime values). We do not explicitly mention this very typical pattern, but the Bias of the flux estimations being always positive for the only-daytime sampling suggests this idea.

The annual cycle behavior of soil respiration is presented in Figure 4-A.

The effect of diurnal and annual variability of environmental variables on the frequency of sampling required for obtaining good performing estimates of soil respiration is discussed in Page 9 Lines 29-32.

**Responses to Anonymous Referee #1**

**General Comments**

This paper is well written and presented. The specific questions the authors address are 1) to assess the performance and accuracy of different numbers of measurements per day 2) compare the performance of linear vs non-linear gap filling based on sampling frequency 3) analyze the effects of including night time respiration measurements on estimations of daily and annual respiration.

I disagree with the authors that questions still remain on how many samples are needed per day; this has been addressed in the literature, which the authors cite. I do agree with the authors that how many samples per year as well as gap filling techniques are issues that need addressed. The authors presented a good examination of issues related to sampling frequencies, gap filling strategies and methods for assessing their performance.

Response: In relation to the question of how many samples are needed per day, we believe this is still an open question. According to the literature review, presented in Table 1, more than half of the studies made in temperate forests only used one sample per day (publication dates range from 1998 to 2015). We explain this in two sentences, Page 2 Line 31 - Page 3 Line 2.

We think a minimum of two samples per day, as in the case of our site, may be more appropriate for other sites as well and, ideally, the number of measurements per day should be assessed at the beginning of every study. We explain this point in Page 10 Lines 20-24.

This manuscript is very useful to researchers planning an effective sampling strategy and contributes to the overall understanding of estimates of daily and annual respiration. I have a couple of issues that the authors should address:

1. The authors used only 3 chambers to represent the "true" soil respiration from their site. In the Davidson et al. 2002 paper that the authors cite, they have a table showing how many soil respiration measurements are needed to be within a certain % of the "true" population mean in a northern temperate forest. Using this as a guideline, the authors, having used only 3 chamber measurements at their site, may only be within ±50% of the true population mean. This is something that I think the authors should address in their discussion. Their manuscript is intended to give guidance as to sampling frequency and so they should also reference how many samples may be necessary to capture the "true" respiration mean per site.

R: We agree with the reviewer. We have a total of five chambers installed in our site, but for two of them we did not have a complete data set. What we meant by true mean was in terms of the temporal variation in soil respiration, and not the spatial variation. We explain this in the Methods section (Page 5 Line 5) and, as suggested by the reviewer, discuss it in the Discussion section (Page 7 Lines 6-8).

2. I understand the use of only the soil temperature model for gap filling since the authors did not see an increase in model fit when adding soil moisture. However, my concern is that there are issues with the soil moisture measurements, the wide range in soil moisture measurements among the 3 respiration chambers is somewhat suspicious. Not including moisture in the gap filling model may have changed the outcome of non-linear gap filling strategy. Can the authors comment on their moisture measurements and the potential impact on their gap filling results? There are also questions regarding the soil moisture measurements below.

R: We respond to this comment in the next question.

**Scientific Questions**

Pg 4 line 29: Did the authors conducted a soil specific calibration for the CS616 probes or use the supplied calibration equation? The bulk density of the soil shows a very wide range in Table 2 and these types of probes do not function as well in soils with low bulk density. Further Figure 4c graph shows a very large range of measured soil water contents among the 3 probes, this might be more related to the calibration equation used than to a true range of soil moisture at the site.

R: The reviewer is right; we had not conducted a previous calibration for the CS616 sensors, although we made gravimetric water content measurements for the three sensors separately. Now we include corrected soil water contents based on the calibrations of the CS616 readings, contrasted with true volumetric water content measurements. These true water content measurements were calculated on the basis of gravimetric water contents and soil bulk density data. As a result of the calibration, the range in soil moisture, as shown in Figure 4C, decreased. We clarified this in the Methods section (Page 4 Lines 28-30).

[Figure]

Figure 4. Daily mean $R_s$ (A), soil temperature (B) and soil water content (C) measured at 5 cm depth. Shaded area represents the minimum and maximum range of three sampling points.

With the corrected values of soil water content, we tested if the non-linear model improved in terms of its performance when adding this variable, not finding new results. The variation of the mean value of soil moisture does not change much during the year (Figure 4C-revised).

Pg 6 line 6: The equation presented here is the Van't Hoff equation: although cited in the Lloyd znd Taylor 1994 paper, it is not their equation.

R: We corrected this (Page 6 Lines 5-6).

Pg 7 line 3: The authors talk about low variability in the cold month and higher in warmer. Can the authors add estimates of the coefficient of variation for these periods?

R: We need to reword this sentence. In fact, if we look at the coefficient of variation of each season (winter 15.9 %, spring 18.3%, summer 18.9%, fall 12.6%), there is not a big difference. We thank the reviewer for the comment that made us realize this and added these estimates of CV in the text (Page 7 Lines 4-6).

Figure 4a: The authors use Rs in $\mu$mols m-2 s-1 in this graph; but use mg in other graphs. It would be preferential to use one type of unit throughout the manuscript. Also note that Figure 4a is missing the $\mu$ in the y axis label.

R: We changed the units in Figure 4A to g $CO_2$ $m^{-2}$ $day^{-1}$ (see Figure 4-revised), so this unit is the same as in Figure 5, where we report the statistics of the estimation of daily Rs. We prefer to keep the unit in kg $m^{-2}$ $yr^{-1}$ in Figure 6 because it is the most commonly used unit for yearly estimates of Rs.

Please add the units for soil water content. These are missing on graphs and in the text.

R: we have added the units of soil water content to Figure 4C (see Figure 4-revised) and in the text, as solicited.

**Technical Correction**

Figure 2: please label the secondary y axis.

R: We corrected Figure 2 (see Figure 2-revised).

[Figure]

Figure 2: Mean long-term (1999–2012) (black) and August 2013-August 2014 (white) monthly precipitation (bars) and air temperature (circles).

Figure 4c: please label the y axis

R: we have added the label to the y axis in Figure 4C ($m^3$ $m^{-3}$).

**Response to Anonymous Referee #2**

The topics raised in this paper are very relevant and informative for researchers planning soil respiration measurements. When comparing CO2 fluxes among different studies it is essential that a standard protocol for measuring daily and annual CO2 is implemented. This manuscript is well written and tackles some of the uncertainties and questions that many researchers encounter when planning field work/ field campaigns. Using an extensive data set, the authors address these specific questions: 1) to assess the performance of estimating daily Rs fluxes based on different numbers of measurements per day 2) to compare the performance of estimating the annual Rs flux using linear vs non-linear interpolation or modelling based on different number of measurements per year and 3) to analyse the effects of including night time respiration measurements on the accuracy of the estimations of daily and annual Rs.

There are a couple of points that should be given some attention:

1) I agree that the issue of daily measurements is still an open question. You mention that most studies take samples in the morning. Could you elaborate for the reader as to why most studies have chosen this time? Also, you conclude that the more appropriate number of samples taken for an effective sampling strategy should be a minimum of two samples per day (one day-time and one night-time). I realise that you propose an assessment at the beginning of each study but could you give a general approximation (time window) as to when those two samples should be taken? For example, should they be taken between 7:00-12:00 and 19:00-23:00?

R: As stated in the manuscript, "Davidson et al. (1998) found that their measurements made between 9:00 AM and 12:00 PM adequately represented the average daily Rs flux in a temperate mixed hardwood forest, although this conclusion was derived from intensive measurements made on only two consecutive days." We found now that Luo and Zhou (2006) made a similar recommendation (making measurements between 9:00-11:00), based on other studies. We believe these suggestions spread out and became a rule of thumb. It could also be due to practical reasons, regarding the use of normal working hours.
Luo, Y. & Zhou, X. (2010). Soil respiration and the environment. Academic Press.

We added a comment about this issue in the Introduction (Page 3 Lines 15-16).

In fact, we found that a minimum of two samples per day (one day-time and one night-time) was an effective strategy. Because we defined 'day' between 7:00 and 19:00, a good way to start would be to take samples between 7:00-10:00 and 19:00-22:00. But, this depends on the latitude of the site, as sunrise and sunset occur at different times of the day. We could add the recommendation to test with the first three hours of the day and the first three hours of the night. However, this should be part of the intensive sampling that we already suggested to do at the beginning of the study.

2) Only 3 chamber measurements were used for this study. Although you are not specifically looking at the spatial variation, I think some discussion needs to be given on the number of samples necessary when advising other researchers on sampling protocols.

R: We think we should not make recommendations on the number of samples to represent the spatial variation, because this is not our focus and other papers have addressed this issue more deeply. However, we added a comment in the Discussion section (Page 7 Lines 6-8) including, as suggested by reviewer #1 the reference to Davidson et al. 2002.

Specific science and written points:

Pg 3 line 9: 'were' instead of 'where' Figure 2 and 4: Simply reiterating the previous reviewer. Make sure the y axis and secondary y axis are labelled correctly and include the soil water units in the text and graphs.
R: All these points have been corrected.

---

## Author Response (AR2)

**Detailed comments:**
P. 1, l .17: Delete "In this aim"; i.e. start with "We randomly selected…"

**R: done.**

P. 1, l. 25: The phrase "In conclusion" comes too soon, as you follow this with further detail of our analysis. I suggest you delete here and may use the phrase for the last sentence (but it's not obligatory there).

**R: "In conclusion" was moved to the last sentence**

P. 2, l. 23: This should be "IRGAs", as the A of IRGA stands for Analyser, so "IRGA analyser" doesn't strictly make sense.

**R: done.**

P. 2, l. 31/32: I propose different punctuation to make this clearer: "In the case of humid forests, both approaches are used, and in the case of annual estimates using manual chambers, measurements are usually done only once per day and during daytime (Table 1)."

**R: done.**

P. 3, l. 3-5: Can I suggest a rephrasing also here to help clarity? "Temperate forests present ecosystems with a high degree of shading compared to ecosystems with sparser vegetation, including agricultural land uses. Nevertheless, variations of ±25% of the daily Rs flux have been reported for a temperate mixed hardwood forest, and mid-morning measurements identified as best suited to estimate daily mean fluxes (Davidson et al., 1998)."

**R: done.**

P. 4, l. 13/14: Annual rainfall should be on the order of 2 m in these systems. Do you mean average daily rainfall here? Or should this in fact be "2,000 – 2,500 mm"? If you state this in metres, this avoids any confusion.

**R: There was a mistake. We changed to point by comma.**

P. 4, l. 23: The depth of collars is an important issue for soil respiration measurements, as a deep insertion can affect root transport of carbohydrates to the rhizosphere. Please state here how deep the insertion of your collars was.

**R: We added the approximate depth (10 cm)**

P. 6, equations 2 and 3: Please check this – your method of calculating standard deviations and RMSE seem identical.

**R: The equations are correct. They are slightly different.**

P. 7, l. 11: Add units to values of soil moisture (2 instances in this line). Also please check the agreement of values stated here and Figure 4. The range shown in panel C of Figure 4 is much higher (in fact too wet to be realistic).

**R: We added the units and corrected the values in the text.**

**The values are OK. We added a sentence to explain that the values are realistic given that these are organic soils. Their water holding capacity is around 0.9, which is explained by the low bulk density and high carbon content.**

Table 1: In the first row, indicate what you mean by period (I presume it's months?), and what M and A means for "Type of Sampling".

**R: We added more information to the headings row.**

I don't normally request that authors cite my work, but I wonder if you consider our efforts to estimate annual soil respiration in a montane forest in Germany as fitting into your category of "humid forest". If yes, you could include it too: (Subke J-A, Reichstein M, Tenhunen JD, 2003. Explaining temporal variation in soil $CO_2$ efflux in a mature spruce forest in Southern Germany. Soil Biology and Biochemistry, 35, 1467-1483.) I don't think there are implications for the text from this inclusion, and it's purely for a more complete representation of studies in the table. Nina Buchmann estimated annual soil respiration using a non-continuous approach in the same stand (but a different year), which could also be included (Buchmann N, 2000. Biotic and abiotic factors controlling soil respiration rates in Picea abies stands Soil Biology and Biochemistry 32, 1625-1635).

**R: we included both citations in Table 1.**